# Anti-Fibronectin Aptamer Modifies Blood Clot Pattern and Stimulates Osteogenesis: An Ex Vivo Study

**DOI:** 10.3390/biomimetics8080582

**Published:** 2023-12-01

**Authors:** Natacha Malu Miranda da Costa, Ludovica Parisi, Benedetta Ghezzi, Lisa Elviri, Sergio Luis Scombatti de Souza, Arthur Belém Novaes, Paulo Tambasco de Oliveira, Guido Maria Macaluso, Daniela Bazan Palioto

**Affiliations:** 1Department of Oral and Maxillofacial Surgery and Periodontology, School of Dentistry of Ribeirão Preto, University of São Paulo, Avenida Do Café-Subsetor Oeste-11 (N-11), Ribeirão Preto 14040-904, SP, Brazil; natachamalu@gmail.com (N.M.M.d.C.); scombatti@forp.usp.br (S.L.S.d.S.); novaesjr@forp.usp.br (A.B.N.J.); 2Laboratory for Oral Molecular Biology, Department of Orthodontics and Dentofacial Orthopedics, University of Bern, Freiburgstrasse 3, 3010 Bern, Switzerland; ludovica.parisi@unibe.ch; 3Centro Universitario di Odontoiatria, Dipartimento di Medicina e Chirurgia, University of Parma, Via Gramsci 14, 43126 Parma, Italy; benedetta.ghezzi@unipr.it; 4Istituto dei Materiali per l’Elettronica ed il Magnetismo, Consiglio Nazionale Delle Ricerche, Parco Area Delle Scienze 37/A, 43124 Parma, Italy; lisa.elviri@unipr.it; 5Department of Basic and Oral Biology, School of Dentistry of Ribeirão Preto, University of São Paulo, Avenida Do Café-Subsetor Oeste-11 (N-11), Ribeirão Preto 14040-904, SP, Brazil; tambasco@usp.br; 6Dipartimento di Scienze Degli Alimenti e del Farmaco, Parco Area Delle Scienze 27/A, 43124 Parma, Italy; guidomaria.macaluso@unipr.it

**Keywords:** aptamer, blood clot, three dimensional, scaffold, hydrogel, osteogenesis

## Abstract

Background: Scaffold (SCA) functionalization with aptamers (APT) provides adsorption of
specific bioactive molecules on biomaterial surfaces. The aim of this study was to observe if SCA
enriched with anti-fibronectin APT can favor coagulum (PhC) and osteoblasts (OSB) differentiation.
Methods: 20 μg of APT was functionalized on SCA by simple adsorption. For PhC formation, SCAs
were inserted into rat calvaria defects for 17 h. Following proper transportation (buffer solution
PB), OSBs (UMR-106 lineage) were seeded over PhC + SCAs with and without APT. Cells and
PhC morphology, PhC cell population, protein labeling and gene expression were observed in
different time points. Results: The APT induced higher alkaline phosphatase and bone sialoprotein
immunolabeling in OSB. Mesenchymal stem cells, leukocytes and lymphocytes cells were detected
more in the APT group than when scaffolds were not functionalized. Additionally, an enriched and
dense fibrin network and different cell types were observed, with more OSB and white blood cells
in PhC formed on SCA with APT. The gene expression showed higher transforming growth factor
beta 1 (TGF-b1) detection in SCA with APT. Conclusions: The SCA functionalization with fibronectin
aptamers may alter key morphological and functional features of blood clot formation, and provides
a selective expression of proteins related to osteo differentiation. Additionally, aptamers increase
TGF-b1 gene expression, which is highly associated with improvements in regenerative therapies.

## 1. Introduction

The use of biomaterials represents one of the main strategies for tissue regeneration [1,2]. In this regard, different kinds of bone substitutes have been developed, and are successfully used in the clinic for the management of bone deficiencies in the oro-/craniofacial area, such as vertical and/or horizontal defects, congenital anomalies (alveolar bone clefts) or challenging defects resulted from traumas, tumor resections and osteonecrosis.

The integration of such biomaterials (scaffold) into the host tissue involves different players, including tissue-resident cells. Once recruited onto the scaffold surface, cells colonize it, proliferate, and differentiate in order to depose and synthesize new specific extracellular matrix (ECM) components, thus restoring the lost tissue integrity and function. However, soon after implantation, biomaterials face blood invasion and thus blood clot formation.

Coagulation involves a series of chemotactic and biomolecular events mediated by different cell types [3,4] and physiologically, its main purpose [5] is the formation of a natural scaffold for the recruitment of cells involved in tissue repair [3,6]. In turn, the new-formed coagulum is the first component to come into contact with biomaterials, as it is the first substrate met by the cells during their recruitment [7]. Considering these premises, it is reasonable to hypothesize that a tune control of blood clot formation may influence cell recruitment at biomaterial interface and thus the whole tissue regeneration.

As we previously mentioned, in recent decades, several three-dimensional (3D) scaffolds have been developed as bone substitutes for the management of critical bone defects [8]. These scaffolds included autografts, allografts, and xenografts. All these materials have been successfully applied in the clinic, but their use often comes with some caveats, such as the morbidity associated with the donor site (autografts) [9] and the risk of cross-infections (allografts and xenografts) [8]. Therefore, synthetic materials have gained considerable popularity since they overcome these issues [10,11]. Furthermore, synthetic materials can be molded on specific-tissue requests with fine control of scaffold geometry, mechanical properties and porosity. This last aspect is of pivotal importance to support homogeneous cellular distribution and the growth of new vessels for the proper distribution of nutrients and oxygen [12,13].

Chitosan is a polysaccharide derived from chitin, one of the main structural components of crustacean exoskeletons, which is among the most studied synthetic biomaterials for the development of 3D scaffolds [14,15,16]. Indeed, chitosan presents excellent biocompatibility, bioresorbability in non-toxic monomers [11,17], antimicrobial properties [18] and capacity to support wound healing [19]. Furthermore, in the recent past, 3D chitosan scaffolds have been successfully printed and have shown to support colonization by human epithelial cells and horse-derived chondrocytes [20]. The association of this hydrogel with beta-tricalcium phosphate (β-TCP) has potentiated osteogenesis in mesenchymal stem cells. In our previous published studies, we also investigated the possibility of further improving chitosan biocompatibility by introducing aptamers on its surface [16].

Aptamers, which are single-strand oligonucleotides with the capacity to recognize and bind potential targets from a great pool of molecules, have been exploited as docking points to promote the selective adsorption of specific ECM molecules on chitosan. This modification resulted in an improved colonization by murine osteoblastic cells (MC3T3) and human epithelial cells (HeLa), as evidenced by the enhancement of focal adhesion expression and the improved organization of cell cytoskeleton [16,21,22].

Additionally, we also observed that when immobilized on other matrices (i.e., hyaluronic acid-based hydrogels), the use of aptamers promoted platelets aggregation and activation, and over the time, enhanced bone regeneration [21]. Hence, we hypothesized that the use of aptamers on 3D chitosan/B-TCP scaffolds could also improve the formation of the blood clot guiding cells’ behavior when in contact with it.

Considering all these premises, the aim of the present work was to evaluate the role of aptamers in controlling coagulum formation. Furthermore, we wish to elucidate how potential differences of the blood clot due to the use of aptamers could affect the behavior of osteoblastic cells.

## 2. Materials and Methods

### 2.1. Ethics Statement

This work was performed in accordance with the Committee of Ethics in Animal Research of the School of Dentistry of Ribeirao Preto, University of Sao Paulo, which approved all the animal procedures performed (process number 2018.1.247.58.7). This article was further written in accordance with the ARRIVE guidelines [23].

### 2.2. 3D Chitosan/B-TCP Scaffolds

A 2% chitosan solution was prepared by dissolving 90% de-acetylated chitosan powder (A.C.E.F., Piacenza, Italy) in a 1% acetic acid solution (Sigma-Aldrich, Saint Louis, MO, USA). D(+)-Raffinose (Sigma-Aldrich) was then added at a final concentration of 290 nM as viscosity modifying agent. ß-tricalcium phosphates granules (ß-TCP) at a final concentration of 20% on the total chitosan solution [24] were finally added to the solution before 3D bioprinting for scaffold manufacturing. The CAS model for 3D printing of scaffold was designed by means of Solidworks^TM^ software (Dassault Systems, Vélizy-Villacoublay, by drawing a grid composed of overlapping parallel filaments set at a nominal distance of 400 μm (Figure 1). Total sizes of the object, composed of five layers, was set to 1.5 × 1.5 × 0.1 cm (width/length/thickness). The 3D printer employed in this study was developed by the Department of Food and Drug Science of the University of Parma [25]. At the end of each manufacturing process, scaffold underwent ionotropic gelation by exposure to saturated ammonia vapors for 1h at room temperature (RT). Scaffolds were subsequently washed by immersion in ddH2O until neutral pH to eliminate ammonia residues and raffinose. Finally, discs (Scaffolds-SCA) of 5 mm were punched and sterilized by immersion in 70% ethanol, transferred to sterile Phosphate Buffer Saline (PBS) until use.

### 2.3. Aptamer Preparation

Commercially available single-stranded DNA aptamers were screened by SELEX technology against human and bovine fibronectin (FN) (ATW0008, Base Pair—Biotechnologies, Pearland, TX, USA). At their 3${}^{\prime }$-end, aptamers were modified with a carbon chain containing a disulfide bond, while with a biotin at their 5${}^{\prime }$-end. Prior to chitosan functionalization, the thiol group at the 3${}^{\prime }$-end was reduced with Reducing Buffer (Base Pair Biotechnologies) for 10 min at RT. Afterwards, excess of the reducing agent was removed with a chromatographic column (mini Quick Spin Oligo columns, Roche Life Science, Penzberg, Bayern, DE) and aptamers diluted at their working concentration in Folding Buffer (Base Pair Biotechnologies).

### 2.4. Scaffold Functionalization: SCA and SCA+APT

At 24 h before aptamer functionalization, SCAs were critically point dried with liquid CO${}_{2}$. Afterwards, SCAs were placed in 96-well plates (Corning, Corning, NY, USA) and incubated with 20 μg reduced aptamers overnight to obtain aptamer-functionalized scaffolds (SCA + APT) [16]. Control scaffolds (SCA) were incubated overnight with a Folding Buffer.

### 2.5. Animals’ Surgery and Physiological Blood Clot Formation

Fourteen 8–12 week old male Wistar Hannover rats weighing 150 g were used in this study (Central Biotherm, USP Campus of Ribeirão Preto).

The surgical procedure was adapted from Monroe et al. (2012) and Zuardi et al. (2023) [26,27]. All animals were anesthetized with an intraperitoneal administration of 10 mg/kg xylazine (Syntec, São Paulo, SP, Brazil) and 80 mg/kg ketamine (Syntec). Afterwards, the surgical side on the skull was shaved off and disinfected with Poly(vinylpyrrolidone).

A superficial incision of 2 cm was operated in order to access the calvaria. Subsequently, using a trephine drill of 5 mm diameters, two bilateral critical size defects were created (5 mm diameter and 2 mm height), and the SCA or the SCA+APT specimens were placed in the same animal following a split mouth approach (Figure 2). Afterwards, wounds were sutured with 5.0 nylon wire (Ethicon, Cincinnati, OH, USA). During surgery, the animal also received a subcutaneous injection of 2 mg/kg tramadol hydrochloride (Cronidor^TM^, Vet Smart, Mannheim, Germany). To reduce the risk of bias, the same trained operator performed all the surgeries.

A total of 24 h after surgeries, the rats were euthanized by anesthetic overdose by administering 0.7 mg/kg of lidocaine and 150 mg/kg sodium thiopental (Syntec). SCAs or SCA + APT samples coated with the respective physiological clots (PhC) were finally harvested and used for further experimental analysis (Figure 2). The ex vivo assay was carried out using 3 biological replicates. In each one, two to four experimental replicates per group per analyses. In total, 42 rats were used, with a mortality rate of 1 rat per biological replicate.

### 2.6. Flow Cytometry (FACS) Analysis

Cell types of the PhC formed on SCAs or SCA+APT samples were assessed by flow cytometry using the BD FACsCanto^TM^ II (BD-Biosciences, East Rutherford, NJ, USA). First of all, cells were purified from the PhC as follows. Samples were gently washed in PBS at RT, treated with 0.25% trypsin (Gibco, Thermo Fisher Scientific, Carlasba, CA, USA) for 5 min at RT, and cells rescued by inhibiting trypsin activity with complete Dulbecco Modified Eagle’s Medium (D-MEM, Gibco) supplemented with 10% Fetal Bovine Serum (FBS). Afterwards, the whole cells were pelleted at 2000 rpm for 5 min, followed by incubation in RBC Lysis Life (eBioscienceTM, Thermo Fisher Scientific) for 10 min at RT for red blood cells lysis. Remaining cells (white blood cells) were thus resuspended in PBS supplemented with 5% FBS, incubated with primary antibody for 30 min on ice, washed 2X in PBS supplemented with 5% FBS, and fixed in 4% paraformaldehyde (PFA) before the analysis. Primary antibodies used for FACS: mouse monoclonal anti-CD45 PE-conjugated (BD-Biosciences), mouse monoclonal anti-CD90 FITC-conjugated (BD-Biosciences), mouse monoclonal anti-CD34 PE-conjugated (Santa Cruz Biotechnology, Dallas, TX, USA), mouse monoclonal anti-CD44 FITC-conjugated (BD-Biosciences), mouse monoclonal anti-CD42 FITC-conjugated (BD-Biosciences), and mouse monoclonal anti-CD61 FITC-conjugated (BD-Biosciences).

Analysis of markers expression was performed by BD FACSDIVA 5.0 software (BD Biosciences) and reported as percentage of cells labeled and mean fluorescence intensity.

### 2.7. Cell Culture

Rat osteoblasts (OSB) UMR-106 were obtained from the American Type Culture Collection (ATCC). Cells were cultured in complete D-MEM supplemented with 10% FBS and 1X Penicillin/Streptomycin (PenStrep, Sigma-Aldrich). Cells were maintained in standard culturing conditions at 37 ${}^{\circ }$C 5% ppCO${}_{2}$. After reaching 70–80% of confluency, cells were collected by 0.25% trypsin and 1 mM EDTA (Gibco, Thermo Fisher Scientific), counted and seeded in 96-well plate on the SCAs or SCA+APT at a final concentration of 5 × 10${}^{4}$ cells/well. Osteogenic commitment was induced 24h after seeding by culturing medium further supplemented with 5 μg/mL ascorbic acid (Sigma-Aldrich) and 7 mM beta-glycerophosphate (Sigma-Aldrich).

### 2.8. Scanning Electron Microscopy

For scanning electron microscopy (SEM) analysis, samples were fixed in 2.5% glutaraldehyde solution (Sigma-Aldrich) in 0.2 M cacodylate buffer (Sigma-Aldrich) for 2 h at RT. Afterwards, samples were dehydrated in increasing ethanol concentrations (30%, 50%, 70%, 90%, 95% and absolute) and sputter coated with a thin layer of gold for 120 s using an SCD 050 coating device (BALTEC, Wallruff, Germany). Image acquisition was performed at 25 kV using a Jeol JSM-6610 LV SEM microscope (Jeol, Peabody, MA, USA).

### 2.9. Immunofluorescence

To evaluate the specific bone-related proteins’ expression, an immunofluorescence staining was performed. For this purpose, 10 days after seeding, UMR-106 was washed with PBS (Gibco, Thermo Fisher Scientific) and fixed with 4% PFA for 10 min at RT. Afterwards, cells were permeabilized with 0.5% Triton x-100 (Sigma-Aldrich) for 10 min, blocked with 5% skimmed milk in PBS (Gibco, Thermo Fisher Scientific) for 30 min, prior to being incubated with primary antibodies in PBS (Gibco, Thermo Fisher Scientific) for 60 min at RT. Cultures were then incubated with secondary fluorescent-labeled secondary antibodies (and anti-mouse Alexa Fluor 488 and anti-rabbit Alexa Fluor 594, both from Molecular Probes, Thermo Fisher Scientific) for 60 min at RT protected from the light. Cell nucleus were counterstained with DAPI (Molecular Probes, Thermo Fisher Scientific) for 5 min at RT, and samples coverslip mounted with the Vectashield^TM^ anti-fade (Vector Laboratories, Burlingame, CA, USA) mounting medium. Samples were examined under a multiphoton microscope image system using the Zeiss LSM 7MP Multiphoton Microscope (Carl Zeiss, Oberkochen, Germany). Acquired images were analyzed with ImageJ (NIH, Bethesda, MD, USA). Primary antibodies used for immunofluorescence: mouse monoclonal anti-bone sialoprotein (IBSP, Developmental Studies Hybridoma Bank, dilution 1:200) and rabbit polyclonal anti-alkaline phosphatase (ALP, Bioss Antibodies, Woburn, MA, USA, dilution 1:100).

### 2.10. RNA Extraction, cDNA Synthesis and Quantitative Real Time Polymerase Chain Reaction (RT-PCR)

Total RNA was isolated using TRIzol^TM^ (Thermo Fisher Scientific) following manufacturer’s recommendation. In brief, 1 mL of TRIzol^TM^ was added to samples for 10 min at RT. After cell lysate collection, samples were transferred to 1.5 mL Eppendorf tubes and 200 μL of chloroform (Sigma-Aldrich) added for 5 min. After centrifugation at 4 ${}^{\circ }$C and 12,000× *g* for 1 min, the upper aqueous phase was collected in new tubes and RNA purified by ethanol 96% (Sigma-Aldrich), followed by further purification with the SVTotal RNA Isolation System kit (Promega, Madison, WI, USA). RNA concentration and purity were measured with a NanoVue PlusTM Spectrophotometer (GE Healthcare, Roche, Basel, Switzerland). A total of 1000 ng of RNA was used as a template for cDNA synthesis through the High-Capacity cDNA Reverse Transcription Kit (Applied Biosystems, Thermo Fisher Scientific) on a Master Cycle Gradient thermocycler (Eppendorf, Hamburg, Germany). Gene expression was detected by real-time polymerase chain reaction (RT-PCR) performed using the TaqMan^TM^ Fast Advanced Master Mix reagent TaqMan probes (Applied Biosystems, Thermo Fisher Scientific) on QuantStudio^TM^ 7 Flex Real-Time PCR (Applied Biosystems, Thermo Fisher Scientific). Data analysis was performed using the dCT method. Osteoblastic marker genes evaluated: alkaline phosphatase (*Alp*), Runt-related transcription factor 2 (*Runx2*), bone gamma-carboxyglutamate protein 2 (*Bglap2*), bone sialoprotein (*Ibsp*) and transforming growth factor beta 1 (*Tgf*β*1*) (Table 1). Glyceraldehyde-3-phosphate dehydrogenase (*Gapdh*) expression was used as endogenous control.

### 2.11. Statistical Analysis

Data obtained were analyzed using Prism 5.0 (GraphPad, Boston, MA, USA). Normal distribution was assessed by Shapiro–Wilk test (*p* < 0.05). To evaluate differences among two groups, Student’s t parametric or Mann–Whitney non-parametric tests were performed. The significance level was set with *p* < 0.05. Results are expressed as mean ± standard deviation.

## 3. Results

### 3.1. Aptamers Promote a Faster Recruitment of Cells Involved in Healing Processes

Electromicrographs show the form and porosity of SCA (Figure 3). Moreover, the OSBs were cultured on SCA functionalized or not with APT. The images demonstrated the OSBs adhered to the SCA filaments (yellow arrow), with no difference in cells number on the different surfaces.

Cells involved in the formation of the physiological coagulum (Figure 4A) were analyzed by FACS for cluster of differentiation (CD) expression (Figure 4B). In particular, mesenchymal stem cells (MSCs—CD90 positive cells), hematopoietic stem cells (HSCs—CD34 positive cells), lymphocytes (CD44 positive cells), leucocytes (CD45 positive cells) and thrombocytes/platelets (CD42 and CD61 positive cells) recruitment were investigated. The cell population selected was similar in each group, varying in size and complexity of cells present in PhC formed on SCA.

The SCA+APT group showed an increased involvement of MSCs (CD90—*p* = 0.0119), but not of the HSCs (CD34—*p* > 0.05). However, even if the amount of HSCs recruited was similar, an increased differentiation of white blood cells was observed in the SCA+APT group compared to SCA (CD44—*p*< 0.0001 and CD45 *p* = 0.0004). On the other hand, all the other markers (CD42 and CD61) spoke for a similar involvement of platelets in both the tested groups (Figure 4C).

To further confirm FACS data, the morphological aspect of blood clots formed on 3D chitosan/β-TCP scaffolds in the presence (SCA + APT) or in the absence (SCA) of aptamers was studied (Figure 4D). An abundance of biconcave cells, most probably erythrocytes, was predominant in the SCA group (Figure 4D, first column, yellow arrowheads). On the contrary, a majority of round cells, assimilable to white cells was observed in the aptamer group (SCA + APT, Figure 4D, second column, yellow arrowheads). This observation corroborates the findings observed in flow cytometry. Interestingly, no spindle-like and elongate cells, referable to MSCs, were observed by SEM in either of the two groups. Taken together, these data suggest the formation of a more mature blood clot in the aptamer group.

### 3.2. The Physiological Blood Clot Formed at Interface of Aptamer-Enriched Scaffold Promote Osteoblasts Differentiation

To investigate the effects potentially triggered by blood clot during bone healing processes, rat osteoblasts UMR-106 were cultured on the physiological clots formed at the interface with the SCA and the SCA + APT group. The rat osteoblastic cells adhesion on the two clots (SCA vs. SCA + APT) appears similar between the two groups, as qualitative evaluated by epifluorescence (Figure 5D, third and fourth column, yellow arrowheads).

Similarly, the clot structure did not influence bone differentiation genes’ expression after 72 h (Figure 6). However, the *Tgf*β*1* expression was more pronounced in the SCA + APT group (Figure 6, *p* = 0.0265), indicating the potential formation of a micro-environment more suitable to bone regeneration. In order to confirm this observation, blood clot formed at the interface of the SCA or of the SCA+APT groups were investigated by immunofluorescence. Interestingly, the expression of osteoblastic markers ALP and IBSP was higher in the SCA + APT group if compared to the control (SCA) (Figure 6, ALP *p* < 0.0001; IBSP *p* = 0.0002).

## 4. Discussion

In this research, the effect of 3D SCA functionalized with aptamers on blood clot formation and, additionally, on osteogenesis was verified. The results showed that the aptamers led to an improvement in PhC patterns, exhibiting greater expression of MSC (CD90) and HSC (CD45 and CD44). The morphological aspect of blood clots formed on chitosan scaffolds with aptamers reveal clear predominance of white blood cells (WBC) in a fibrin mesh enriched with ECM. In the same group, greater ALP and IBS protein labeling and *Tgf*β*1* gene expression was associated with osteoblastic differentiation.

The use of 3D chitosan/β-TCP based SCA in this investigation was carried out because this biomaterial is highly hydrophilic, with excellent biological properties [28] for 3D SCA elaboration for bone regeneration [14,15,16]. In this context, the 3D biomaterial’s functionalization has been proposed, aiming the adsorption of specific molecules on the surface, in an attempt to modify its bioactivity and physiology [29]. Inspired by natural healing hematoma, tissue-engineered SCA/hydrogels and prefabrication strategies attract significant interests in developing functional bone biomaterial [30].

Here, anti-fibronectin aptamers functionalization was proposed because the FN is a glycoprotein widely available in blood plasma, especially helping in platelet aggregation and fibrinogenesis [31,32]. On the other hand, FN will reinforce fibrin mesh formation that will serve for progenitor’s cells recruitment [33]. In addition, this important ECM protein is indispensable in many physiological processes [34,35], being known to participate in early stages of osteogenesis [36,37].

The use of blood clots formed in an animal model allowed its PhC formation. This method reflects the simultaneous orchestration of immune response along with cellular interactions of the host’s clotting. There are ex vivo models of blood clot formation, where blood is collected and immediately added to SCA [38]. However, the proper cascade of events that happens in vivo is underestimated. In this study, PhC was created in the rat’s calvaria defects, allowing the release of substantial growth factors (GFs), MSC and immune cells recruitment and, after that, transported to analysis. With this, we sought to observe the whole process in the coagulum formation mimicking a prohealing microenvironment. It was observed that aptamers induced MSC (CD90) prevalence in PhC. MSC can differentiate into specific cell types [39] and lead to secretion of various cytokines and chemokines that contribute to OSB differentiation [40]. This paracrine signaling can influence resident cells, promoting immunomodulatory [41], anti-apoptotic and antioxidant [42] effects. Thus, the MSC expressive detection in PhC formed on SCA with APT may, in addition to recruit cell types associated with tissue repair, potentiate the release of substantial GFs for osteoimmunology.

It was also observed a predominance of lymphocytes (CD45) and leukocytes (CD44) in PhC with APT. CD45 is a tyrosine phosphatase receptor protein, an essential regulator of T and B cells, protagonists in innate immune system mediation [43]. The CD44 is a glycosaminoglycan that participates in cellular interactions with ECM, such as adhesion, motility and proliferation [44]. Leukocytes are cells normally present in fibrin clots, being a significant source of cytokines and GFs, and can act synergistically with platelets present in blood clots [45]. In addition, some studies report the role of CD44 in ECM stiffness [44]. Together with various cell types (including MSC), lymphocytes and leukocytes participate in the healing process.

The morphology results corroborate FACS results. The morphology of PhC formed on aptamer functionalized on SCA showed a blood clot enriched with a fibrin mesh composed mainly of different WBC types, which could be blood, MSC or immune cells. As seen in the 3D view by multiphoton microscopy, the OSB distribution on PhC formed in SCA with APT showed to be denser and with more cells when compared to the SCA group. This will create a cellular environment conducive to expression of genes associated with osteogenesis, including favoring Tgfβ1 stimulation.

The functionalization with aptamers resulted in high *Tgf*β*1* gene expression observed in RT-PCR assay. Tgfβ1 is a chemotactic polypeptide for macrophages and fibroblasts [46], in addition to having high affinity with ECM proteins, such as FN [47]. It supports deposition of main ECM components [48] directly influencing cell differentiation being important in osteogenesis signaling pathway [49]. Additionally, leukocytes and platelets, cells normally attached to the fibrin meshwork, are rich sources of GFs [50]. Thus, *Tgf*β*1* expression is indirectly related to WBC adherence and was increased in PhC formed on SCA functionalized with anti-fibronectin aptamers. This fact possibly leads to *Tgf*β*1* releases from leukocytes and platelets, acting directly on ECM deposition and osteodifferentiation.

Another point to be emphasized is that the groups did not show quantitative differences in platelets detection. In flow cytometry, CD42 and CD61 was used to verify the platelets’ presence in PhC formed on SCA. The results revealed no difference among the SCA and SCA + APT groups. Despite the strong association between *Tgf*β*1* and leukocytes [51], there is evidence showing the relationship between CD45 [52] and CD44 [53] in platelet activation. In our analysis, there was CD45 and CD44 prevalence in PhC formed on SCA functionalized with APT. Despite the fact that there are no quantitative differences in platelets’ detection, the major detection of leukocytes and lymphocytes in PhC formed on SCA+APT group can possibly lead to an improvement in platelet activation, which can directly influence biological behavior of blood clot.

In osteodifferentiation, ALP is fundamentally known as an activator of initiating matrix mineralization mechanisms [54]. This metalloenzyme is easily found on cell surface and in vesicle matrix of some tissues [55]. When activated, it will induce hydroxyapatite (HA) deposition in vesicles and these will cross cell membranes, releasing HA into ECM [56]. The HA adhesion to collagen and maturation of this matrix will be assisted by proteins present in ECM, such as BSP. This anionic phosphoprotein will mediate OSB behavior and consolidate mineralized matrix deposition to collagen [57]. The immunofluorescence performed in this research showed higher ALP and BSP immunolabeling for the SCA + APT group.

Despite the limitations of this research, it is possible to infer that the enrichment of a 3D scaffold with anti-fibronectin aptamer may alter key morphological and functional features of blood clot formation, favoring osteoblastic behavior in ex vivo assays, amplifying positive effects on osteodifferentiation. The results of this research show new perspectives on the role of aptamer not only in osteogenesis, but also in immunoinflammatory response associated with tissue repair. The aptamer anti-fibronectin adsorption on 3D biomaterials shows promising features to osteoimmunomodulation and healing microenvironment.

## 5. Conclusions

The anti-fibronectin aptamer adsorption on 3D biomaterials shows promising features to osteoimmunomodulation and healing microenvironment.

## Figures and Tables

**Figure 1 biomimetics-08-00582-f001:**
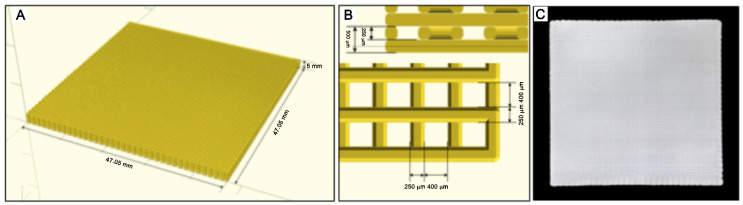
(**A**)—CAD image created in Solidworks^TM^ software for SCA printing. (**B**)—The SCA porosity established in the CAD image. (**C**)—SCA image after 3D printing.

**Figure 2 biomimetics-08-00582-f002:**
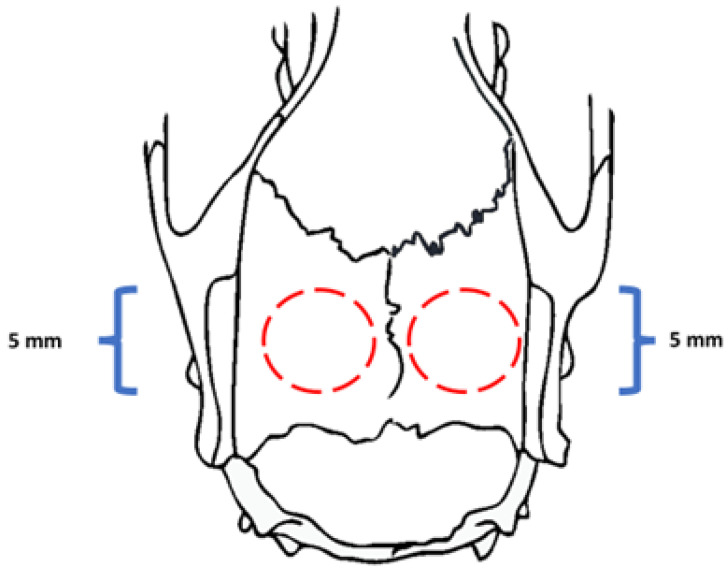
Scheme showing the exact position where the critical size bone defect was created in rat calvaria with a trephine drill to physiological clot assay (red dashed circle). The 5 mm diameter defect was created bilaterally in the center of each parietal bone rat skull.

**Figure 3 biomimetics-08-00582-f003:**
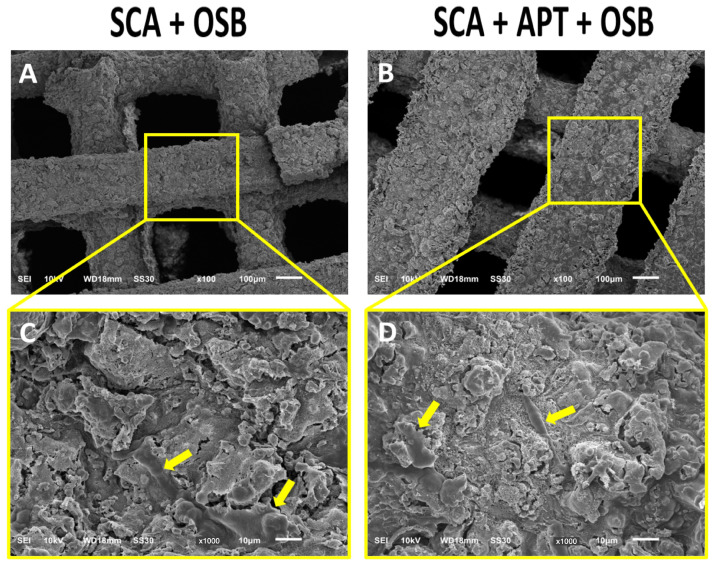
Electromicrographs showing SCA+OSB (**A**,**C**) and SCA+APT+OSB (**B**,**D**), arrows (**C**,**D**) showing the OSB cells on SCA filaments. The image shows the OSB adhered to the SCA filaments (yellow arrow), with no difference in cells number on the different surfaces. Scale: 10 μm and 100 μm.

**Figure 4 biomimetics-08-00582-f004:**
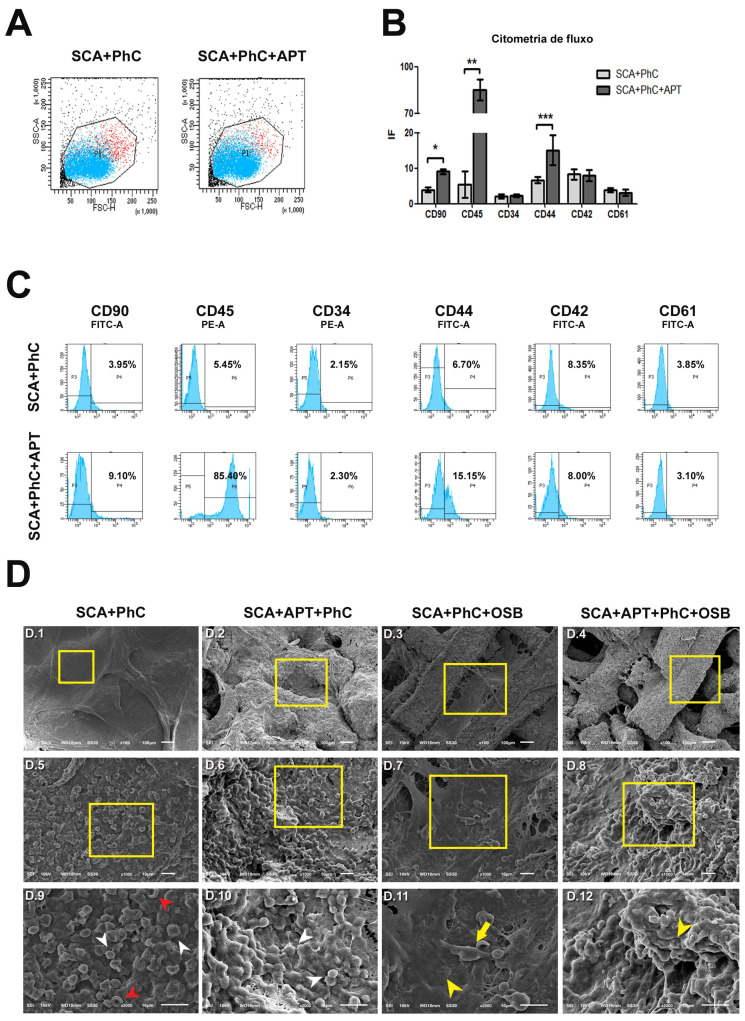
(**A**)—Flow cytometry cell population on the experimental groups, showing the cell population selected in PhC formed on SCA was similar in each group, varying in size and complexity of cells; (**B**)—graph showing higher expression of CD90 (*), CD45 (**) and CD44 (***) in SCA + APT + PhC when compared to SCA+PhC; (**C**)—histograms showing the average of cells labeled for CD90, CD45, CD34, CD44, CD42 and CD61 among the groups; (**D**)—electromicrographs showing SCA + PhC (**D.1**,**D.5**,**D.9**), SCA + APT + PhC (**D.2**,**D.6**,**D.10**), SCA + PhC + OSB (**D.3**,**D.7**,**D.11**) and SCA + APT + PhC + OSB (**D.4**,**D.8**,**D.12**) groups at 100× (**D.1**–**D.4**), 1000× (**D.5**–**D.8**) and 2000× (**D.9**–**D.12**) magnification. Red and white arrowheads are showing the red blood cells and white blood cells, respectively (**D.9**,**D.10**). Yellow arrow (**D.11**) and arrowheads (**D.11**,**D.12**) are showing different cell types on PhC formed on SCAs. Scale bars: 100 μm (**D.1**–**D.4**) and 10 μm (**D.5**–**D.12**). Significance value: * *p* < 0.05, ** *p* < 0.01, *** *p* < 0.001.

**Figure 5 biomimetics-08-00582-f005:**
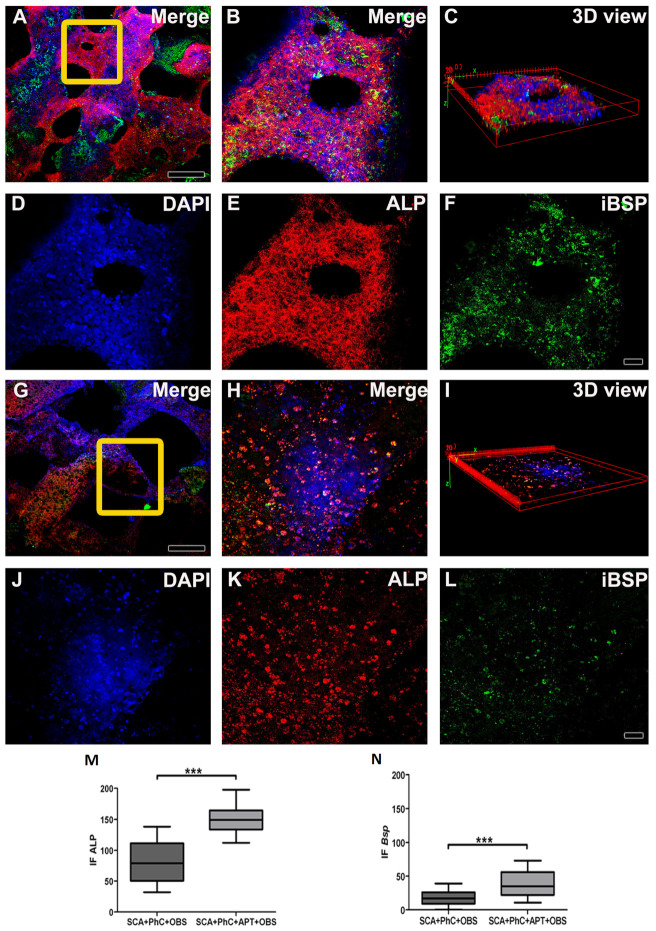
Immunofluorescence images obtained at Multiphoton Microscopy showing OSB on PhC formatted on SCA functionalized (**A**–**F**) and non-functionalized (**G**–**L**) with APT. A 3D image formed of OSB and PhC formed on SCAs with (**C**) and without APT (**I**). The cell nucleus was stained with DAPI (**D**,**J**), ALP with Alexa Fluor 594 (**E**,**K**) and BSP with Alexa Fluor 488 (**F**,**L**). ALP (**M**) and BSP (**N**) immunoexpression in the SCA + ATP + PhC + OSB group was higher when compared to SCA + PhC + OSB group. Scale: 20 μm and 50 μm and. Significance value: *** *p* < 0.001.

**Figure 6 biomimetics-08-00582-f006:**
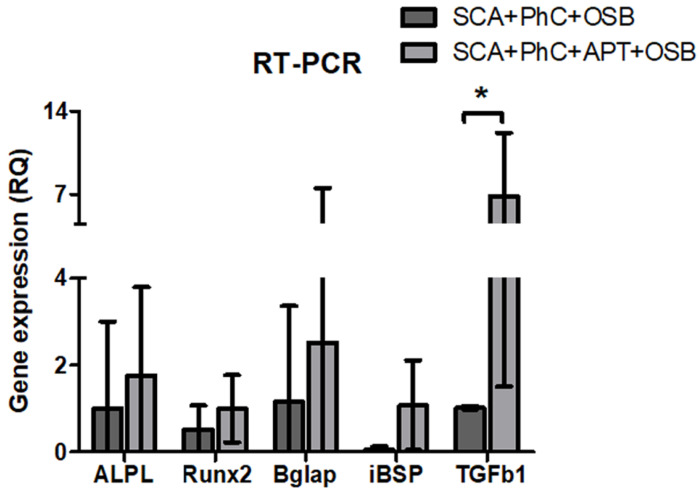
Graphs showing RT-PCR results, with differences in *Tgfβ1* gene expression (*). Significance value: * *p* < 0.05.

**Table 1 biomimetics-08-00582-t001:** TaqMan probes (Thermo Fisher Scientific) used in RT-PCR assay.

Gene’s Symbol	Gene’s Names	TaqMan Assay
*Runx2*	Runt related transcription factor 2	Rn01512298_m1
*Alp*	Alkaline phosphatase liver/bone/kidney	Rn01516028_m1
*Bsp*	Bone sialoprotein	Rn00561414_m1
*Bglap*	Bone Protein Containing Gamma-carboxyglutamic acid (Osteocalcin/oc)	Rn00566386_g1
*Tgf* β *1*	Transforming growth factor beta 1	Rn00572010_m1
*Gapdh*	Glyceraldehyde-3-phosphate dehydrogenase	Rn99999916_s1

## Data Availability

The data presented in this study are available on request from the corresponding author under plausible justification.

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
