# Peer review of "Anti-Fibronectin Aptamer Modifies Blood Clot Pattern and Stimulates Osteogenesis: An Ex Vivo Study"

_biomimetics, 2023, doi:10.3390/biomimetics8080582_

Round 1
Reviewer 1 Report (Previous Reviewer 2)
Comments and Suggestions for Authors
Comments addressed
Comments on the Quality of English LanguageNA
Author Response
Reviewer 1
Comment: Moderate editing of English language required.
Answer: English revision will be carried out in the final review of the manuscript.
Reviewer 2 Report (New Reviewer)
Comments and Suggestions for Authors
The objective of this research on the functionalization of Scaffold (SCA) with aptamers (APT) to adsorb specific bioactive molecules on biomaterials surface. The study investigated the effect of anti-fibronectin APT enriched SCA on coagulum (PhC) and osteoblasts (OSB) differentiation. The author team showed that SCA functionalization with APT provides a selective expression of proteins related to coagulation pattern and osteo differentiation, and also increase TGF-β1 gene expression, which is highly associated with improvements in regenerative therapies.
The research outline is well-defined and the methods used in this study are robust. The research outline is well-defined and the methods used in this study are robust. The reported data and results are appropriate and clearly explained. However, based on the reviewer's opinion, there is a major concern regarding the main storyline of the draft: SCA+aptamers lead to blood clot formation and promote osteogenesis. The reviewer has pointed out that blood clot/coagulum formation is strongly correlated with platelet adhesion and accumulation, and there was no significant change in platelet accumulation between the SCA+PhC and SCA+PhC+APT groups in this work. Furthermore, Fig 3 confirms that the OSB adhered to both groups, with no difference in cell numbers (although the image data may have some inaccuracy). Therefore, it appears that SCT+APT will not impact coagulation patterns and osteodifferentiation
However, it is worth noting that in a different story line: SCA+APT did bring significant changes to white blood cell differentiation, and the expression of Tgfβ1 was more pronounced in this group, indicating a favorable environment for bone regeneration. Thus, it is suggested that adding APT will lead to WBC differentiation and favor Tgfβ1 stimulation. Blood clot/coagulum formation should not play a significant role in this comparison (with and without APT).
In the next round of revision, the author team should focus on addressing the relationship between blood clots/coagulum and white blood cells, and how the latter impact the formation of the former. It is important to note that blood clots are strongly related to platelets rather than white blood cells, which is against the reviewer's expertise. By emphasizing the significant changes observed in white blood cell differentiation and Tgfβ1 expression, the author team can strengthen their argument for the inclusion of APT in the study.
Besides this major concern, the reviewer listed other questions that arose from reading the paper. The authors are encouraged to address these questions in the revised work. Good luck!
· Line 39: Autografts may not always be the best option during surgery as the patient's own tissue may not have sufficient time to act as proper scaffolds. Can you provide a reference to support your citation of autografts?
· Section 2.2: Consider including both the CAD image of the designed scaffold and the actual entity as a new figure in this draft.
· Line 156: Typo, 5X 104
· Line 232 and 234: Please highlight the different red and white blood cells using distinct colors and refer to Figure 2D9 or 2D10.
· Line 244: It would be helpful to clarify whether the similarity in adhesion of rat osteoblasts on the two clots refers to their morphology or amount. Stating specifically which aspect is similar would provide more clarity, as simply saying "adhesion was similar" could be interpreted in various ways.
· Line 246: How do you characterize the cell metabolic activity in Figure 2B?
· Line 258: Consider specifying the white blood cells as HSC, as you have used CD34 to mark HSC and noted no significant change in line 224.
· Line 259: Can you provide additional data to support the claim that the morphological aspect indicates a clear predominance of WBC? It seems that the related data, Figure 2D9 and 2D10, do not provide sufficient evidence to support this conclusion. Could you please address this concern?
· Line 314: Based on the evidence you provided, it appears that Tgfβ1 is released from leukocytes, but its release is not strongly related to platelets
· Line 325 and you conclusion: SCA+APT did modify something, such as WBC differentiation, but it is unlikely to affect the formation patterns of coagulation. That’s the core concern in the whole manuscript.
Comments on the Quality of English LanguageMinor editing of English language required
Author Response
Reviewer 2
Comment: Q1: Line 39: Autografts may not always be the best option during surgery as the patient's own tissue may not have sufficient time to act as proper scaffolds. Can you provide a reference to support your citation of autografts?
Answer: We agree with reviewer. The reference was added accordingly and is highlighted in the text.
Alayan, J.; Ivanovski, S. A prospective controlled trial comparing xenograft/autogenous bone and collagen-stabilized xenograft for maxillary sinus augmentation—Complications, patient-reported outcomes and volumetric analysis. Clinical Oral Implants Research 2018, 29, 248–262.
Comment: Q2: Section 2.2: Consider including both the CAD image of the designed scaffold and the actual entity as a new figure in this draft.
Answer: The reviewer has a point. The CAD image of the designed scaffold and an SEM image of the actual scaffold are now presented as new Figure 1.

Figure 1. A - CAD image created in Solidworks™ software for SCA printing. B - The SCA porosity established in the CAD image. C – SCA image after 3D printing.
Comment: Q3: Line 156: Typo, 5X 104
Answer: The indicated typo has been corrected and is highlighted in the text.
Comment: Q4: Line 232 and 234: Please highlight the different red and white blood cells using distinct colors and refer to Figure 2D9 or 2D10.
Answer: We agree with the reviewer. Red and white arrows were inserted to show red blood cells and white blood cells, respectively.

Figure 4. A - Flow cytometry cell population on the experimental groups, showing the cell population selected in PhC formed on SCA was similar in each group, varying in size and complexity of cells; B -Graph showing higher expression of CD90 (*), CD45 (**) and CD44 (***) in SCA+APT+PhC when compared to SCA+PhC; C- Histograms showing the average of cells labelled for CD90, CD45, CD34, CD44, CD42 and CD61 among the groups; D- Electromicrographs showing SCA+PhC (D.1, D.5, D.9), SCA+APT+PhC (D.2, D.6, D.10), SCA+PhC+OSB (D.3. D.7, D.11) and SCA+APT+PhC+OSB (D.4, D.8, D.12) groups at 100X (D.1-D.4), 1000X (D.5 – D.8) and 2000X (D.9 – D.12) magnification. Red and white arrowheads are showing the red blood cells and white blood cells, respectively (D.9 and D.10). Yellow arrow (D.11) and arrowheads (D.11 and D.12) are showing different cell types on PhC formed on SCA’s. Scale bars: 100 µm (D.1 – D.4) and 10 µm (D.5 – D.12). Significance value: * p <0.05, ** p <0.01, *** p <0.001.
Comment: Q5: Line 244: It would be helpful to clarify whether the similarity in adhesion of rat osteoblasts on the two clots refers to their morphology or amount. Stating specifically which aspect is similar would provide more clarity, as simply saying "adhesion was similar" could be interpreted in various ways.
Answer: The similarity in adhesion of rat osteoblasts on the two clots refers to their amounts, as qualitative evaluated, readily visualized by epifluorescence. The text has been modified as follows:
“The rat osteoblastic cells adhesion on the two clots (SCA vs. SCA+APT) appears similar between the two groups, as qualitative evaluated by epifluorescence (Figure 2D, third and fourth column, yellow arrow heads).”
Comment: Q6: Line 246: How do you characterize the cell metabolic activity in Figure 2B?
Answer: We apologize for the mistake in the edition of the final text. The cell metabolic activity has not been included in the present manuscript. This parameter will further be explored in a future project.
Comment: Q7: Line 258: Consider specifying the white blood cells as HSC, as you have used CD34 to mark HSC and noted no significant change in line 224.
Answer: We agree with the reviewer. We made a mistake in the manuscript when referring to hematopoietic progenitor cells. CD34 was used to detected hematopoietic progenitor cells, with no difference between the groups evaluated. Otherwise, CD45 and CD44 markers were used to verify leukocytes and lymphocytes cells, respectively. The results showed a prevalence of CD45 and CD44 in blood clot formed on SCA+APT group. The paper has been modified (as described below) and these changes are highlighted in yellow in Results and Discussion section.
“In particular mesenchymal stem cells (MSCs - CD90 positive cells), hematopoietic stem cells (HSCs - CD34 positive cells), lymphocytes (CD44 positive cells), leucocytes (CD45 positive cells) and thrombocytes/platelets (CD42 and CD61 positive cells) recruitment were investigated.”
Comment: Q8: Line 259: Can you provide additional data to support the claim that the morphological aspect indicates a clear predominance of WBC? It seems that the related data, Figure 2D9 and 2D10, do not provide sufficient evidence to support this conclusion. Could you please address this concern?
Answer: The SEM images are the only images we obtained to verify the blood clot morphology formed on SCA in different experimental groups. New arrows have been added for a more accurate visualization of WBC localization (images 2D9 and 2D10).
Comment: Q9: Line 314: Based on the evidence you provided, it appears that Tgfβ1 is released from leukocytes, but its release is not strongly related to platelets.
Answer: Platelets are the primary mediators of hemostasias and thrombosis, and are thus key elements in blood clot formation. In flow cytometry, CD42 and CD61 were used to verify the presence of platelets in the blood clot formed on SCA in different experimental groups. In the Results, no differences were observed between the SCA and SCA+APT groups. Despite the strong association between Tgfb1 and leukocytes (Torres et al., 2022), there is evidence showing the relationship between the presence of CD45 (Inamdar et al., 2019) and CD44 (Liu et al., 2016) in platelet activation. In our analysis, there was a prevalence of CD45 and CD44 in blood clot formed on SCA functionalized with APT. Despite there are no quantitative differences in platelets detection, the prevalence of leukocytes and lymphocytes in the blood clot formed on SCA+APT group can lead to an improvement in platelet activation, which can directly positively influence blood clot physiology. These data evidence a possible association between APT and improvement the biological behaviour of blood clot in SCA. This description was added to the paper discussion.
“Another point to be emphasized is that the groups did not show quantitative differences in platelets detection. In flow cytometry, CD42 and CD61 was used to verify the platelets presence in PhC formed on SCA. The results revealed no difference among the SCA and SCA+APT groups. Despite the strong association between Tgfβ1 and leukocytes [51], there is evidence showing the relationship between CD45 [52] and CD44 [53] in platelet activation. In our analysis, there was CD45 and CD44 prevalence in PhC formed on SCA functionalized with APT. Despite there are no quantitative differences in platelets detection, the major detection of leukocytes and lymphocytes in PhC formed on SCA+APT group can possibly lead to an improvement in platelet activation, which can directly positively influence biological behaviour of blood clot.”
Torres, L.S.; Chweih, H.; Fabris, F.C.; Gotardo, E.M.; Leonardo, F.C.; Saad, S.T.O.; Costa, F.F.; Conran, N. TGF-β1 Reduces Neutrophil Adhesion and Prevents Acute Vaso-Occlusive Processes in Sickle Cell Disease Mice. Cells 2022, 11, 1200.
Inamdar, V.V.; Kostyak, J.C.; Badolia, R.; Dangelmaier, C.A.; Manne, B.K.; Patel, A.; Kim, S.; Kunapuli, S.P. Impaired glycoprotein VI-mediated signaling and platelet functional responses in CD45 knockout mice. Thrombosis and Haemostasis 2019, 119, 1321–1331.
Liu, G.; Liu, G.; Alzoubi, K.; Chatterjee, M.; Walker, B.; Muenzer, P.; Luo, D.; Umbach, A.T.; Elvira, B.; Chen, H.; et al. CD44 sensitivity of platelet activation, membrane scrambling and adhesion under high arterial shear rates. Thrombosis and Haemostasis 2016, 115, 99–108
Comment: Q10: Line 325 and you conclusion: SCA+APT did modify something, such as WBC differentiation, but it is unlikely to affect the formation patterns of coagulation. That’s the core concern in the whole manuscript.
Answer: As verified by flow cytometry, there was a prevalence of CD45 and CD44 in SCA+APT group. The leukocytes and lymphocytes in blood clot formed on SCA functionalized with APT may possibly lead to an improvement in platelet activation, which may directly influence the biological behaviour of blood clot. These data directly demonstrate the relationship between APT and the improvement in coagulation patterns on SCA.
Reviewer 3 Report (New Reviewer)
Comments and Suggestions for Authors
It seems that the manuscript was submitted along with the comments from the previous reviewers. It can clearly see the huge improvements. However, there are still rooms for the improvement, from my point of view. My comments are below:
· The resolution of both Figure 2A and Figure 3 are too low;
· The author shall use labels, for example the dashed-box, to indicate which parts were enlarged, in Figure 2D;
· It was mentioned in the article that “with no difference in cells number on the different surfaces”, how would the author have such conclusion?
· The author need thorough check the writing thoroughly (such as Electrofotomicrographs, SEM electro micrographs showing, et al.);
· The authors need to check the format of the manuscript, for example, the indent in the section of Discussion.
Comments on the Quality of English LanguageSuch as Electrofotomicrographs, SEM electro micrographs showing, et al.
Author Response
Reviewer 3
Comment: Q1: The resolution of both Figure 2A and Figure 3 are too low;
Answer: Another image with better resolution was inserted to the manuscript.
Comment: Q2: The author shall use labels, for example the dashed-box, to indicate which parts were enlarged, in Figure 2D;

Figure 4. A - Flow cytometry cell population on the experimental groups, showing the cell population selected in PhC formed on SCA was similar in each group, varying in size and complexity of cells; B -Graph showing higher expression of CD90 (*), CD45 (**) and CD44 (***) in SCA+APT+PhC when compared to SCA+PhC; C- Histograms showing the average of cells labelled for CD90, CD45, CD34, CD44, CD42 and CD61 among the groups; D- Electromicrographs showing SCA+PhC (D.1, D.5, D.9), SCA+APT+PhC (D.2, D.6, D.10), SCA+PhC+OSB (D.3. D.7, D.11) and SCA+APT+PhC+OSB (D.4, D.8, D.12) groups at 100X (D.1-D.4), 1000X (D.5 – D.8) and 2000X (D.9 – D.12) magnification. Red and white arrowheads are showing the red blood cells and white blood cells, respectively (D.9 and D.10). Yellow arrow (D.11) and arrowheads (D.11 and D.12) are showing different cell types on PhC formed on SCA’s. Scale bars: 100 µm (D.1 – D.4) and 10 µm (D.5 – D.12). Significance value: * p <0.05, ** p <0.01, *** p <0.001.
Answer: Yellow dashed boxes were inserted in Figures 2D.1- 2D.8.
Comment: Q3: It was mentioned in the article that “with no difference in cells number on the different surfaces”, how would the author have such conclusion?
Answer: The similarity in adhesion of rat osteoblasts on the two clots refers to their amounts, as qualitative evaluated, readily visualized by epifluorescence. The text has been modified as follows:
“The rat osteoblastic cells adhesion on the two clots (SCA vs. SCA+APT) appears similar between the two groups, as qualitative evaluated by epifluorescence (Figure 2D, third and fourth column, yellow arrow heads).”
Comment: Q4: The author need thorough check the writing thoroughly (such as Electrofotomicrographs, SEM electro micrographs showing, et al.);
Answer: We agree with the reviewer. The text has been modified in the paper accordingly.
Comment: Q5: The authors need to check the format of the manuscript, for example, the indent in the section of Discussion.
Answer: The format of all manuscript was checked.
Round 2
Reviewer 3 Report (New Reviewer)
Comments and Suggestions for Authors The comments were well addressed, and the manuscript can be accepted as it is.This manuscript is a resubmission of an earlier submission. The following is a list of the peer review reports and author responses from that submission.
Round 1
Reviewer 1 Report
Comments and Suggestions for Authors
1. Please add a few more references in the introduction, specifically where biomaterials other than chitosan are mentioned. The introduction is one long and convoluted paragraph, please split in at least two, to improve reading and comprehension.
2. It would be very helpful if the authors added a schematic to the manuscript that clearly shows where the animal implantation was done.
3. The authors should add the number of animals that were used, as well as their mortality rate during the experimental assay.
4. A few references would immensely strengthen the cell morphological analysis portion of the manuscript, as it appears to be an important distinction determinant between cell types.
5. A more in-depth analysis of the cell markers that are impacted for the cells that were collected from the clots would be important to better characterize them, and to show if other gene pathways, such as inflammation, angiogenesis, or cell death, are also impacted.
6. As the anti fibronectin aptamers was not bound to the scaffold, a release graph would be very helpful to show how fast APT diffuses out of the scaffold and how stable the SCA-APT complex is over time.
7. Higher magnification of the Figure 3 immunofluorescence images would be very useful to better appreciate the results the authors are presenting.
Comments on the Quality of English Language
1. The abstract is very heavily loaded with acronyms, including a couple (ALP & BSP) that are not explained. This makes it very difficult to get into the paper, so please reduce the number of acronyms and abbreviations for the abstract
2. The discussion is hard to read, it would be very helpful if it was split into several, shorter paragraphs.
Reviewer 2 Report
Comments and Suggestions for Authors
For most part, the paper is well-conceived and well-written. The following are some changes to improve the quality of the manuscript.
Several recent works that 3D printed/ bioprinted chitosan and have not been cited (Line 48). These papers need to be cited,
- Ramesh, S., Kovelakuntla, V., Meyer, A. S., & Rivero, I. V. (2021). Three-dimensional printing of stimuli-responsive hydrogel with antibacterial activity. Bioprinting, 24, e00106.
Xu, J., Zhang, M., Du, W., Zhao, J., Ling, G., & Zhang, P. (2022). Chitosan-based high-strength supramolecular hydrogels for 3D bioprinting. International Journal of Biological Macromolecules.
Gwak, M. A., Lee, S. J., Lee, D., Park, S. A., & Park, W. H. (2023). Highly gallol-substituted, rapidly self-crosslinkable, and robust chitosan hydrogel for 3D bioprinting. International Journal of Biological Macromolecules, 227, 493-504.
CoÅŸkun, S., Akbulut, S. O., Sarıkaya, B., Çakmak, S., & GümüÅŸderelioÄŸlu, M. (2022). Formulation of chitosan and chitosan-nanoHAp bioinks and investigation of printability with optimized bioprinting parameters. International Journal of Biological Macromolecules, 222, 1453-1464.
Authors could consider including pictures of the printed scaffolds.
Comments on the Quality of English LanguageNA
Reviewer 3 Report
Comments and Suggestions for Authors
The authors performed a complete study on the in vivo behavior of chitosan/b-tcp scaffolds functionalized with aptamers. The results are very interesting and can be published after major revision.
1. Paragraph 2.2: I am confused about the final shape of the scaffolds and the reason for the 3d printing process. Although 3d printed scaffolds in the form of grids are printed, the authors “punch the scaffolds in the form of disks” before the implantation in animals
2. There are no results and comments on the 3d printed scaffolds. For example the authors have to show optical or SEM images of the scaffolds and comment the printability of the chitosan/b-tcp system
3. Enrich the “conclusions” paragraph
4. What kind of 3D printer was used?
5. Paragraph 2.2: How was elimination of raffinose was evaluated?